# *OLR1* Is a Pan-Cancer Prognostic and Immunotherapeutic Predictor Associated with EMT and Cuproptosis in HNSCC

**DOI:** 10.3390/ijms241612904

**Published:** 2023-08-17

**Authors:** Lei Wu, Yuantong Liu, Weiwei Deng, Tianfu Wu, Linlin Bu, Lei Chen

**Affiliations:** State Key Laboratory of Oral & Maxillofacial Reconstruction and Regeneration, Key Laboratory of Oral Biomedicine Ministry of Education, Hubei Key Laboratory of Stomatology, School & Hospital of Stomatology, Wuhan University, Wuhan 430079, China

**Keywords:** *OLR1*, prognosis, immunotherapy, EMT, cuproptosis

## Abstract

Metabolism plays a critical role in cancer. *OLR1* has been implicated in cardiovascular and metabolic disorders, while its association with tumorigenesis and tumor immunity remains poorly defined in the literature. We conducted comprehensive pan-cancer analyses based on the TCGA database to examine *OLR1* expression and its prognostic implications. Correlations between *OLR1* expression level and tumor immunity and immunotherapy were investigated by immune infiltration, enrichment, and TIDE analysis methods. Immunohistochemistry detected *OLR1* expression in HNSCC. We used the GSEA method to explore the potential signaling pathways in which *OLR1* is involved, and a correlation analysis to investigate the relationships between *OLR1* and epithelial–mesenchymal transition (EMT) and cuproptosis. In addition, the effects of *OLR1* knockdown on the EMT process, invasion, stemness, and cuproptosis of HNSCC cells were examined by scratch, Transwell, CCK8, sphere formation, and flow cytometry, while changes in related proteins were detected using the immunoblotting method. *OLR1* is highly expressed in most cancers, and it is associated with patient prognosis. *OLR1* expression positively correlates with immunosuppressive cell infiltration and immune checkpoint molecules, while being negatively associated with effector T cells. Moreover, significant correlations are observed between *OLR1* expression and tumor mutation burden (TMB) and microsatellite instability (MSI) in some cancers. In HNSCC, *OLR1* expression is related to advanced clinicopathological factors and unfavorable outcomes. Patients with high *OLR1* expression levels are prone to experience immune escape and benefit less from immune checkpoint inhibitor (ICI) therapy. Moreover, *OLR1* expression may affect EMT, stemness, and cuproptosis resistance outcomes. *OLR1* is an immune-related prognostic biomarker with potential as a prognostic indicator for immunotherapy, and it may also be involved in regulating the EMT process and cuproptosis in HNSCC.

## 1. Introduction

In recent decades, cancer has become the leading cause of death, with an estimated 19.3 million new cancer cases and nearly 10.0 million cancer deaths occurring in 2020, and cancer is a significant barrier to improving life expectancy worldwide [1]. Although the number of cancer survivors continues to increase in developed countries, mainly due to advances in early prevention and treatment modalities, the mortality rate of cancer patients worldwide still shows a remarkable increasing trend [2,3]. Immunotherapy, exemplified by immune checkpoint inhibitor (ICI) therapy, is gradually becoming a therapeutic standard in the field of cancer treatment planning, in addition to radiotherapy and chemotherapy [4,5]. Nonetheless, only a minority of patients benefit from it, and low treatment response rates and drug resistance are the greatest obstacles to the development of such therapies [6]. Therefore, further research is urgently required to verify the efficacy of cancer immunotherapy in diverse cancer types [7]. The epithelial–mesenchymal transition (EMT) process and cancer stem cells are widely regarded in the literature to be the main culprits in tumor initiation, recurrence, and metastasis [8], exerting a substantial influence on the efficacy of ICI therapy [9]. Cuproptosis is a recently identified form of programmed cell death that is closely related to cellular metabolism [10,11]. However, in the field of tumor immunity, fewer reports exist concerning the role of cuproptosis. Consequently, further empirical evidence is required to substantiate its implications in this domain. 

As the research field evolves, scientific attention has been increasingly focusing on the common characteristics present across different human malignancies to better understand the underlying mechanisms of tumor development [12]. A pan-cancer analysis, which examines the genes of multiple cancers and compares the differences in gene expression, has been widely used in the field to identify tumor molecular markers and signaling pathways. This, in turn, has improved our understanding of the molecular mechanisms underlying tumor initiation and development processes [13,14].

Emerging research has revealed a strong connection between metabolic disorders and tumor development processes [15]. Oxidized low-density lipoprotein receptor 1 (*OLR1*), the receptor of oxidized low-density lipoprotein (ox-LDL), is mainly expressed in endothelial cells, macrophages, smooth muscle cells, platelets, fibroblasts, and neurons [16]. The effects of *OLR1* have been studied in metabolic and cardiovascular diseases [17]; however, it is also overexpressed in some cancers and involved in various tumorigenic processes. In breast cancer, upregulated *OLR1* predicts poor prognosis and promotes migration [18,19]. In pancreatic cancer, *OLR1* promotes metastasis through c-MYC and HMGA2 [20]. Additionally, in colon cancer, *OLR1* overexpression drives glycolytic metabolism, leading to proliferation and chemoresistance [21]. In addition, *OLR1* has been implicated in the progression of esophageal cancer through autophagy [22,23]. Lectin-like oxidized low-density lipoprotein receptor-1 (LOX-1), a translated protein of *OLR1*, has also been implicated in cancer [24]. In pancreatic cancer, LOX-1 is associated with EMT and metastasis processes [25], whereas LOX-1 knockdown can suppress colorectal cancer progression and metastasis [26]. Moreover, LOX-1 is one of the most overexpressed proteins in PMN-MDSC, and LOX-1^+^ PMN-MDSC presents potent immunosuppressive activity [27,28]. However, the precise role of *OLR1* in different tumor types, including HNSCC, and its relevance to tumor immunity and immunotherapy remain unclear in the research.

In this study, we used multiple public databases to evaluate *OLR1* expression and its relationship with prognosis in 33 cancer types, and to explore correlations between *OLR1* expression levels and immune infiltration and immune checkpoint expression. Finally, a validation was performed in HNSCC, and the correlations of *OLR1* expression levels with the EMT process, stemness, and cuproptosis were investigated. The present study aims to evaluate the potential of *OLR1* as a promising biomarker for predicting the prognosis and treatment efficacy of ICI, while also highlighting its association with the EMT process and cuproptosis resistance in HNSCC. This study provides new insights to improve immunotherapy response rates.

## 2. Results

### 2.1. Aberrant Expression and Prognostic Value of OLR1 in Pan-Cancer

To determine the significance of *OLR1* in malignancies, *OLR1* mRNA expression levels were initially scrutinized in 33 cancer types, using the comprehensive dataset obtained from The Cancer Genome Atlas (TCGA). The results show that *OLR1* is markedly upregulated in the majority of cancers, with a few exceptions. Intriguingly, *OLR1* expression was downregulated in DLBC, LUAD, LUSC, SARC, and TGCT compared to normal tissues. (Figure 1A). Furthermore, the paired comparative analysis showed that *OLR1* expression was significantly higher in BRCA, CHOL, COAD, ESCA, HNSCC, KIRC, KIRP, PRAD, STAD, and THCA compared to adjacent normal tissues. In contrast, a reduced expression of *OLR1* was observed in tumor tissues of LUAD and LUSC (Figure 1B).

To investigate the prognostic significance of *OLR1* expression across 33 different types of cancer, we performed a survival analysis using a Cox proportional hazards model. Our results indicated that patients with elevated *OLR1* expression levels experienced worse OS in CESC, COAD, HNSCC, LGG, SARC, STAD, THCA, and THYM. Conversely, in CHOL, LUAD, and SKCM, patients with high *OLR1* expression levels had a better prognosis (Figure 2A). This observation was further validated by the Kaplan–Meier survival analysis (Figure 2B–K). Subsequently, we examined the correlations between *OLR1* expression and PFI and DSS. Our analysis demonstrated that elevated *OLR1* expression was closely associated with poor PFI in ACC, COAD, GBM, HNSCC, LGG, LUSC, PAAD, and PRAD; however, in BLCA, CHOL, DLBC, and SKCM, a high *OLR1* expression level represented a better PFI outcome (Appendix A). Similarly, the univariate Cox regression analysis of DSS outcomes indicated that high *OLR1* expression was a risk factor for DSS in patients with ACC, COAD, ESCA, HNSCC, LGG, LUSC, PAAD, and UCEC; however, the opposite result was observed in CHOL, KIRP, LUAD, PCPG, and SKCM (Appendix A). Collectively, our results suggest that *OLR1* expression levels are significantly elevated in the progression of various cancers and have the potential to serve as a prospective pan-cancer prognostic biomarker, particularly for COAD, HNSCC, LGG, and CHOL. 

### 2.2. Immune Infiltration Analysis of OLR1 in Pan-Cancer

Increasing evidence suggests that cancer metabolism is not only involved in regulating tumor initiation and development, but also leads to the reprogramming of immune cell metabolic pathways, resulting in impaired anti-tumor immune responses [29]. At present, targeting cancer metabolism has been shown in the research to synergistically enhance immunotherapy [30]. Thus, we investigated the association between the metabolic gene *OLR1* and tumor immunity in our study, aiming to ascertain its potential as a therapeutic target in cancer immunotherapy. The tumor microenvironment (TME) plays a crucial role in tumorigenesis and progression [31]. In an immunosuppressed TME, tumor-infiltrating immune cells are dysfunctional, resulting in the evasion of immune surveillance and the unbridled proliferation of malignant cells [32]. A series of algorithms, namely, CIBERSORT, EPIC, TIMER, XCELL, MCPCOUNTER, and QUANTISEQ, were used to assess the correlation between *OLR1* expression and the level of immune cell infiltration. The results of these algorithms show that *OLR1* expression is positively correlated with several cell types, particularly macrophages, myeloid dendritic cells, regulatory T cells, and cancer-associated fibroblasts, while it is negatively correlated with some specific T-cell subtypes, such as naive and Th1 CD4+ T cells (Figure 3 and Appendix A).

Immune checkpoints have emerged as a vital mechanism that is exploited by tumor cells to evade recognition and attack by T cells. ICI therapy has, to date, proven to be a clinically effective intervention for various cancers [4,5]. To this end, we investigated the correlations between *OLR1* and 17 common immune checkpoint expressions in pan-cancer. Notably, except for DLBC and TGCT, *OLR1* exhibited positive correlations with most immune checkpoint molecules in almost all cancers, with a particularly strong correlation with HAVCR2 (Figure 4A). TMB and MSI are closely correlated with the efficacy of ICI therapy [33]. It was found that the expression level of *OLR1* displayed strong correlations with MSI and TMB in diverse cancers (Figure 4B,C). In particular, *OLR1* exhibited a positive correlation with TMB in BLCA, COAD, PRAD, SARC, and THYM, but a negative correlation in LIHC, LUAD, LUSC, and PAAD. In addition, *OLR1* showed a positive correlation with MSI in COAD and READ; but conversely, it exhibited a negative correlation with MSI in HNSCC, LGG, LUSC, OV, SKCM, and STAD. Together, the expression level of *OLR1* in various cancers is closely associated with the infiltration of multiple immune cells and can be used as a marker of suppressive tumor immune response, and it is expected to serve as a predictor of ICI response.

### 2.3. Enrichment Analysis of OLR1-Related Genes in Pan-Cancer

To further elucidate the molecular mechanism of *OLR1* in tumorigenesis, we performed gene ontology (GO) and Kyoto Encyclopedia of Genes and Genomes (KEGG) functional enrichment analyses on the top 100 proteins interacting with *OLR1* obtained from the STRING database (Appendix A and Appendix A). The GO analysis was divided into three parts as follows: The biological processes (BP) enriched in this dataset were mainly related to “response to molecule of bacterial”, “response to lipopolysaccharide”, “regulation of inflammatory” and “regulation of cell–cell adhesion”; the cellular components (CC) enriched were related to “external side of plasma membrane”, “endocytic vesicle”, and “tertiary granule membrane”; and the enriched molecular functions (MF) were linked to “lipoprotein particle binding”, “cargo receptor activity”, and “scavenger receptor activity” (Figure 5A). According to the KEGG analysis, we unexpectedly observed that *OLR1* was closely associated with “the PD-L1 expression and PD-1 checkpoint pathway in cancer”, “the NOD-like receptor signaling pathway”, “TNF signaling pathway”, and “the Toll-like receptor signaling pathway” (Figure 5B), suggesting that *OLR1* may play a role in tumor immune regulation. We also found a close link between *OLR1* and HIF-1 as well as the NF-κB signaling pathway, suggesting that *OLR1* may play an important role in the processes of tumorigenesis and metastasis. In addition, the GO and KEGG analyses both suggested that *OLR1* was associated with intercellular adhesion, suggesting that it may be involved in regulating cell migration. Furthermore, we constructed a protein–protein interaction (PPI) network for *OLR1* and the top 10 associated genes using STRING (Figure 5C). Additionally, we found that most of these genes were reported to be associated with tumor progression. We also investigated whether *OLR1* affected antitumor drug sensitivity using the CellMiner dataset [34] and found that *OLR1* correlated with resistance to drugs, such as vincristine and paclitaxel (Figure 5D, Appendix A). These results suggest that *OLR1* expression may impact the efficacy of chemotherapy in certain cancers.

### 2.4. OLR1 Overexpression in HNSCC and Resistance to Immunotherapy 

The abovementioned results indicate that *OLR1* may play an important role in HNSCC. Considering the fact that immunotherapy exhibits a response rate of less than 20% in HNSCC patients, there is an urgent need to develop novel targets or strategies to enhance the efficacy of immunotherapy [35]. Therefore, as a follow-up to our pan-cancer analysis, we further investigated the underlying mechanisms by which *OLR1* influences HNSCC progression, the sixth most common cancer worldwide [1]. First, we observed a significant increase in *OLR1* expression levels in tumor tissues compared to the normal controls in the TCGA-HNSCC dataset (*p* < 0.001, Figure 6A,B). Then, we analyzed the correlation between *OLR1* expression and clinical features, demonstrating that HNSCC patients with higher T stage (TNM classification) had higher *OLR1* expression levels than those with lower T stage (T1 vs. T3 and T4, *p* < 0.05, Figure 6C). Additionally, *OLR1* expression was positively correlated with a higher pathological grade (I vs. II, *p* < 0.01; I vs. III and IV, *p* < 0.01, Figure 6D). Then, we performed IHC staining and confirmed that LOX-1 was overexpressed in both human and mouse HNSCC (*p* < 0.05 and *p* < 0.01, respectively, Figure 6E,F). Notably, both the univariate and multifactorial Cox regression analyses indicated that *OLR1* could serve as an independent prognostic factor for OS in HNSCC patients (*p* = 0.004 and *p* = 0.013, respectively, Figure 6G). Furthermore, the Kaplan–Meier survival analysis using the log-rank test in the GSE41613 datasets reinforced our findings, suggesting that HNSCC patients with elevated *OLR1* expression levels presented worse outcomes (*p* = 0.01, Figure 6H). Therefore, these results suggest that *OLR1* overexpression is closely associated with aggressive cancer behavior and poor prognosis in HNSCC.

Then, we used the tumor immune dysfunction and exclusion (TIDE) algorithm to evaluate the potential clinical efficacy of immunotherapy in different *OLR1*-expressing subgroups. Our results reveal that *OLR1* expression is significantly reduced in HNSCC patients exhibiting a high efficacy of immunotherapy responses compared to those who are immunotherapy resistant (Figure 6I). Additionally, the TIDE scores were markedly elevated in the *OLR1* high-expression subgroup in contrast to the *OLR1* low-expression subgroup, implying that HNSCC patients with higher *OLR1* expression levels were more likely to be resistant to ICI therapy than those with lower *OLR1* expression levels (*p* < 0.001, Figure 6J). Moreover, we found that the MSI score was higher in the *OLR1* low-expression subgroup (*p* < 0.001, Figure 6K), while the T-cell exclusion and T-cell dysfunction scores were higher in the *OLR1* high-expression subgroup (*p* < 0.05 and *p* < 0.001, respectively, Figure 6L,M). These results suggest that *OLR1* expression is associated with cancer immune escape and targeting *OLR1* may increase tumor T-cell infiltration as well as MSI, thereby improving the response rate to ICI therapy.

### 2.5. OLR1 Knockdown Inhibits the EMT Process and Stem Cell Properties via the STAT3 Pathway in HNSCC Cells

Given the association between *OLR1* expression and clinical outcomes in HNSCC, we subsequently sought to explore the potential mechanisms by which *OLR1* contributes to HNSCC progression. A gene set enrichment analysis (GSEA) of *OLR1* was performed using the TCGA-HNSCC dataset and we found that *OLR1* may be involved in multiple oncogenic signaling pathways; notably, there was a strong positive correlation between *OLR1* and the EMT process (Figure 7A). In addition, *OLR1* showed a significant positive correlation with mesenchymal markers (*FN1*, *VIM*, and *MMP9*) and EMT-TFs (*SNAI1* and *ZEB1*, Figure 7B). The EMT process has been widely demonstrated to be associated with invasion, metastasis, immunosuppression, and therapy resistance in a variety of tumors [8]. Thus, to further verify whether *OLR1* affected the EMT process in HNSCC, we utilized siRNA to reduce *OLR1* expression in the HNSCC cell lines Cal27 and SCC4. The Western blotting results show that the “mesenchymal” characteristics (N-cadherin and Vimentin) of Cal27 and SCC4 diminish following *OLR1* deletion, while the “epithelial” characteristic (E-cadherin) is significantly enhanced (Figure 7C). This result suggests that *OLR1* silencing restricts epithelial–mesenchymal plasticity in HNSCC. Subsequently, through the cell scratch assay, we found that the healing ability of *OLR1* knockdown cells was significantly inhibited (*p* < 0.001, Figure 7D). Moreover, we used the Transwell experiment to verify that the invasive ability of Cal27 cells was markedly suppressed following the *OLR1* knockdown (*p* < 0.01, Figure 7E). 

Next, we investigated the regulatory mechanism of *OLR1* in HNSCC. The results of the GSEA analysis suggest that *OLR1* is associated with STAT3 signaling (Figure 7A), which is an important signaling pathway for invasion and the maintenance of stemness properties [36]. Consistent with this result, we observed that *OLR1* knockdown significantly reduced STAT3 activation by tyrosine phosphorylation and the expression of stemness marker (CD44) (Figure 7F). Furthermore, our study also showed that the knockdown of *OLR1* inhibited the proliferative capacity and sphere-forming ability of HNSCC cells (*p* < 0.01 and *p* < 0.05, respectively, Figure 8A,B). Furthermore, the ALDEFLUOR assay revealed that the proportion of ALDH^high^ cells was decreased in the si*OLR1* group, suggesting the critical role of *OLR1* in regulating stemness properties in HNSCC (*p* < 0.01 and *p* < 0.05, respectively, Figure 8C,D). Taken together, these data suggest roles for *OLR1* in the maintenance of the EMT process and stemness characteristics through the STAT3 pathway in HNSCC.

### 2.6. OLR1 Acts as a Repressor of Cuproptosis in HNSCC Cells

Cuproptosis, which is a unique mode of cell death, is highly correlated with cellular metabolism [11]. Cells dependent on glycolysis tend to be more resistant to cuproptosis than cells dependent on mitochondrial respiration, suggesting that targeting glycolysis may be effective in enhancing the efficacy of cuproptosis [37]. Considering that *OLR1* can drive glycolytic metabolism [21], we decided to further explore the relationship between *OLR1* and cuproptosis sensitivity. Therefore, we subsequently integrated the TCGA-HNSCC database and observed correlations between *OLR1* and known cuproptosis-related genes [10] (*FDX1*, *DLAT*, *ATP7A*, *ATP7B*, *CDKN2A*, and *GLS*) (Figure 9A). Therefore, we decided to explore the relationship between *OLR1* and cuproptosis. Elesclomol acts as a copper ionophore and exerts antitumor effects by effectively inducing the onset of cuproptosis [38]. In our study, we found that elesclomol-Cu promoted the death of HNSCC cells in a concentration-dependent manner (Figure 9B). Importantly, the knockdown of *OLR1* significantly enhanced elesclomol-Cu-induced cell death (Figure 9B). This finding was further verified using the annexin V/PI assay. In addition, tetrathiomolybdate (TTM) prevented cell death (*p* < 0.001, Figure 9C,D). In the process of cuproptosis, copper binds to lipoylated DLAT and induces the oligomerization of DLAT. The increase in insoluble DLAT leads to cytotoxicity and induces cuproptosis [10]. We found that, following elesclomol-Cu treatment, the *OLR1* silencing group presented a more pronounced DLAT oligomerization compared with the control group (Figure 9C,D). Together, these results suggest that *OLR1* may inhibit the occurrence of cuproptosis by inhibiting the oligomerization of DLAT.

## 3. Discussion

The role of *OLR1* in tumorigenesis has only recently come to light, with previous studies primarily focusing on cardiovascular and metabolic diseases. Recent reports in the field have identified its overexpression in diverse malignancies, including breast [18,19], colon [21], esophageal [22,23], and pancreatic cancers [20,25]. These results are consistent with our results. We validated this outcome in HNSCC and found that *OLR1* expression was significantly higher in HNSCC and correlated with advanced clinicopathological factors in the TCGA dataset. In addition, *OLR1* has been closely correlated with clinical outcomes in a few cancer types [19,20,25], which was also observed in our results. Specifically, both univariate and multifactorial survival analyses indicated that *OLR1* was a poor prognostic indicator for HNSCC patients. These results suggest that *OLR1* may be a valuable prognostic biomarker across different cancer types.

The TME consists of diverse immune cell types that can play a crucial role in either tumor cell clearance or tumor immune escape [39,40]. Previous studies have revealed that the degree of immune cell infiltration in the TME is related to tumor development and prognosis [41]. This study determined that *OLR1* expression was positively correlated with several cell types in a variety of tumors, particularly macrophages, myeloid dendritic cells, regulatory T cells, and cancer-associated fibroblasts, which play major roles in promoting angiogenesis, immunosuppression, metastasis formation, and therapy-resistance behaviors. There is evidence that the aberrant activation of *OLR1* in PMN-MDSCs is associated with tumor immunosuppression [27,28]. Therefore, we preliminarily speculated that *OLR1* might promote tumor development by recruiting these immunosuppressive cells to infiltrate the TME.

In recent years, immunotherapies based on ICI have developed rapidly. They have achieved impressive results in the treatment of several types of cancer. However, only a small percentage of patients have experienced clinical benefits [4]. Therefore, there is a need for the development of biomarkers that can predict whether patients will benefit from these therapies. Immune checkpoints, such as CTLA-4, PD-1, and TIM-3, play important roles in regulating T-cell responses and have been shown to be effective targets for cancer therapy [42,43]. The expression of constitutive co-suppressor receptors on T cells inhibits effector T-cell responses. Notably, our correlation analysis results showed that *OLR1* was positively correlated with most immune checkpoint molecules in the majority of cancer types, with a particularly strong correlation with *HAVCR2*. The KEGG analysis also revealed a significant correlation between *OLR1* and the PD-1/PD-L1 pathway. To further investigate the relationship between *OLR1* and immunotherapy, we correlated *OLR1* expression with TMB and MSI, which were evaluated as potential biomarkers for predicting ICI therapy responses [33]. Additionally, we found that *OLR1* was correlated with both TMB and MSI in specific cancer types. Moreover, TIDE integrates expression signatures of T-cell dysfunction and T-cell exclusion to mimic tumor immune evasion, and thus predict clinical responses to ICI [44]. An elevated TIDE score correlates with an increased tendency for immune evasion and a reduced response to ICI therapy. Consequently, the TIDE algorithm was used to validate the relevance of *OLR1* to immunotherapy. Interestingly, the results show that HNSCC patients with high *OLR1* expression levels have higher TIDE, T-cell exclusion, and dysfunction scores, but lower MSI scores. These results suggest that patients with high *OLR1* expression levels may be resistant to treatment with ICI. Taken together, *OLR1* may serve as a potential predictor for immunotherapy. 

EMT is an important process of normal embryonic development and tissue regeneration processes [45]. However, abnormal reactivation of the EMT process during cancer progression and metastasis is associated with malignant properties of tumor cells, including promoting migration and aggressiveness, as well as increasing tumor stemness [8]. Recent studies have revealed that the EMT process can modulate the tumor microenvironment and facilitate immune escape, resulting in immunotherapy resistance [46]. *OLR1* has been demonstrated to participate in modulation of the EMT process, thereby instigating the metastasis of pancreatic cancer [20,25] and osteosarcomas [47]. In line with these findings, we observed a robust positive correlation between *OLR1* and common EMT-related genes in pan-cancer (Appendix A). Moreover, the GSEA results showed that *OLR1* exhibited the strongest correlation with the EMT hallmark in HNSCC. The correlation analysis results further verified that *OLR1* was positively correlated with mesenchymal markers and EMT-TFs. Thus, to investigate whether *OLR1* affected the EMT phenotype, we knocked down the expression of LOX-1 in vitro. The Western blotting results showed that the mesenchymal properties of HNSCC cells weakened after *OLR1* depletion. Additionally, the knockdown of *OLR1* significantly inhibited the invasive ability of HNSCC cells. The EMT process was closely related to stem cell-like properties, and we also found suppressed stemness and proliferation abilities in the si*OLR1* group. The STAT3 signaling pathway was implicated in the regulation of proliferation, survival, invasion, stem cells, and immunosuppression [36]. Furthermore, its interaction with the EMT process has been identified as a crucial regulator of tumor metastasis [48]. Notably, our study revealed that inhibiting *OLR1* impeded activation of the STAT3 pathway. These results suggest that *OLR1* may play a pivotal role in modulating the EMT process via the STAT3 pathway in HNSCC.

Copper nanoparticle-based therapeutics have demonstrated the ability to induce cuproptosis in cancer cells, thereby notably enhancing dendritic cell maturation and promoting infiltration by antitumor CD8+ T cells [49]. An increasing number of researchers, by employing bioinformatics analyses, have demonstrated that cuproptosis was implicated in tumorigenesis, further contributing to the intricate complexities of tumor immune evasion [50]. This evidence suggests a potential association of cuproptosis with immunogenic phenotypes. Exploring cuproptosis holds the potential to improve our understanding of tumorigenesis and tumor microenvironment remodeling behaviors. At present, limited research exists regarding a correlation between HNSCC and cuproptosis, with only one bioinformatics study indicating a potential correlation between cuproptosis and prognosis, as well as immune responses in HNSCC [51]. Considering the important role of *OLR1* in driving tumor glycolysis and its relevance to the immunosuppressive tumor microenvironment and tumor metastasis, we further investigated the relationship between *OLR1* and cuproptosis. Our results showed that *OLR1* silencing significantly enhanced elesclomol-induced cuproptosis. Copper directly binds to DLAT, an enzyme involved in the regulation of the mitochondrial tricarboxylic acid (TCA) cycle, promoting the disulfide-bond-dependent aggregation of lipoylated DLAT, leading to cellular cuproptosis. Consistent with this notion, *OLR1* knockdown enhanced DLAT oligomerization following elesclomol treatment. These results suggest that *OLR1* may negatively regulate elesclomol by regulating the oligomerization of DLAT. 

Our study results revealed the potential significance of *OLR1* in tumor immunity and prognosis; however, there are still some limitations worth considering. Most of the conclusions of this study were based on bioinformatics analysis, and while we conducted in vitro experiments to support our hypothesis, further in vivo experiments on mice would provide a more complete understanding of the immunomodulatory effects of *OLR1*. Our results suggest a potential association between *OLR1* and pro-immunogenic tumor phenotypes; however, further mechanistic studies are required to elucidate the underlying molecular pathways, as well as clinical validations to determine its utility in guiding treatment decisions. Another limitation of this study was the relatively small sample size of the clinical cohort that we used, which may have affected the statistical power of our analysis. Despite these limitations, our study sets the stage for future research in this area and provides valuable insights into the potential role of *OLR1* in tumor therapy.

## 4. Materials and Methods

### 4.1. Bioinformatics Data and Resources 

RNA-seq data for 546 HNSCC cases including 502 HNSCC tumor samples and 44 normal tissue samples, were obtained from The Cancer Genome Atlas (TCGA) database (https://portal.gdc.cancer.gov/, accessed on 7 March 2023). RNA expression data and survival information for 97 HNSCC samples (GSE41613) were obtained from the Gene Expression Omnibus (GEO; https://www.ncbi.nlm.nih.gov/geo/, accessed on 10 March 2023) [52]. All raw data are provided in the Appendix A. 

In order to assess *OLR1* expression patterns across various cancer types, we retrieved RNA-Seq data from the TCGA database. The TCGA data portal was searched to obtain transcripts per million (TPM) values for *OLR1* in different cancer types. The TPM value is a normalization method used for RNA-seq, which considers both sequencing depth and gene length. In our study, we focused on tumor samples and utilized Log2(TPM+1) values. A differential expression analysis was performed using t-tests or ANOVA statistical tests based on the distribution of the data. The significance threshold was set at *p* < 0.05 after correction for multiple testing. To visualize the expression pattern, boxplots and heatmaps were generated using R-based data visualization packages.

### 4.2. Survival Analysis

Kaplan–Meier and Cox regression analyses were conducted to explore the influence of *OLR1* on patient prognosis using the R packages “survminer” and “survival”. Overall survival (OS), progression-free interval (PFI), and disease-specific survival (DSS) values were also evaluated. The hazard ratio (HR) for deaths associated with *OLR1* expression was estimated by using the univariate Cox proportional hazards regression model. Then, a multivariate Cox model was constructed to estimate the adjusted HR for *OLR1* expression; *p*-values less than 0.05 were considered to be statistically significant.

### 4.3. PPI, GO, and KEGG Analyses

The protein–protein interaction (PPI) network and enrichment analysis and the STRING (https://string-db.org/, accessed on 8 August 2023) database were used to construct the PPI of *OLR1*. The minimum required interaction score was set at 0.4. The gene oncology (GO) and Kyoto Encyclopedia of Genes and Genomes (KEGG) analyses of OLR1-related molecules were performed using clusterProfiler packages.

### 4.4. Immune Landscape Analysis

The TIMER2.0 database (http://timer.comp-genomics.org/, accessed on 27 March 2023) provided 6 different immune infiltrating cell algorithms: CIBERSORT, EPIC, TIMER, XCELL, MCPCOUNTER, and QUANTISEQ [53]. The scoring data of the 6 tumor-infiltrating immune cells were downloaded from the TIMER2.0 database. Spearman’s correlation test was used to investigate the relationship between *OLR1* expression and the 6 types of immune cells in pan-cancer. In addition, the TIMER2.0 database was used to assess the correlations between immune checkpoint-related and EMT-related gene expressions and *OLR1* expression.

To predict the immunotherapy response in different *OLR1* subgroups, tumor immune dysfunction and exclusion (TIDE) scores for 502 HNSCC samples were calculated using an algorithm provided by Jiang et al. [44]. The TIDE score evaluated two distinct tumor immune evasion mechanisms: the dysfunction of tumor-infiltrating cytotoxic T lymphocytes (CTLs) and the exclusion of CTLs by immunosuppressive factors.

Level 4 simple nucleoside variation data processed with MuTect2 in GATK V4.1.8.1 software (Broad Institute, Cambridge, MA, USA) were acquired from the TCGA database, and the TMB was calculated [54]. The microsatellite instability signatures for 33 tumors were obtained from reported studies [55]. Correlations between *OLR1* mRNA expression and MSI or TMB were investigated by Spearman’s correlation test.

### 4.5. Gene Set Enrichment Analysis (GSEA)

For the signaling pathway analysis, the HNSCC patients were initially divided into two subgroups based on the median expression levels of *OLR1*. Then, a differential expression analysis was performed between the high- and low-expression subgroups. The input genes of GSEA were sorted by their log fold change (logFC) values. Then, an enrichment analysis was performed using the GSEA method with the cluster Profiler package of R to identify the signaling pathways in which the differentially expressed genes were involved [56].

### 4.6. HNSCC Samples

The human tissue samples analyzed in our study included 6 primary HNSCC and 5 normal oral mucosae collected from the School and Hospital of Stomatology, Wuhan University, Department of Oral Maxillofacial-Head Neck Oncology, in 2022, with informed consent obtained from the patients. All human tissue samples were collected with reference to our previous studies [57]. Age, sex, tumor grading, and TNM staging were obtained from patient medical records (Appendix A). The 4-nitroquinoline-1-oxide (4NQO)-induced HNSCC specimens used in this study were collected and preserved as paraffin-embedded blocks in our previous study [58].

### 4.7. Immunohistochemistry

Immunohistochemical staining was conducted as previously described [59]. Briefly, human and mouse HNSCC samples were paraffin embedded and sectioned at 4 μm. Following xylene deparaffinization and gradient ethanol soaking rehydration, the sections were subjected to high-temperature antigen retrieval with a citrate antigen retrieval solution and endogenous peroxidase inhibition with 3% hydrogen peroxide. Then, the non-specific binding was blocked using 5% goat serum (Mxb Biotechnologies, Fuzhou, China, KIT-9710). The processed sections were then incubated with primary antibodies (Appendix A) at 4 °C overnight, washed with PBS, and incubated with horseradish peroxidase-conjugated secondary antibodies (Mxb Biotechnologies, KIT-9710) at 37 °C for 30 min. A DAB substrate kit (Mxb Biotechnologies, DAB-0031) and hematoxylin were used for staining. Images were visualized using a Pannoramic MIDI scanner (3DHISTECH) and quantified using the Aperio ScanScope software (v11.1.2.752, Aperio, San Diego, CA, USA). IgG was used as a negative control.

### 4.8. Cell Lines

The Cal27 and SCC4 cells were obtained from the China Center for Type Culture Collection (Shanghai, China) and were grown for no more than 15 generations. The Cal27 cells were cultured in high glucose DMEM, while the SCC4 cells were cultured in F12/DMEM. Mycoplasma contamination assays were performed annually by PCR.

### 4.9. Western Blotting

The experiments were performed as previously described [59]. The extraction of total protein was performed with RIPA buffer supplemented with protease and phosphatase inhibitors (Roche, P7626; P2850). A BCA protein assay kit (Beyotime, Shanghai, China, P0012S) was used to determine the protein concentrations. The protein samples were separated on a 10% SDS-PAGE gel and transferred onto a polyvinylidene difluoride membrane (Roche, Basel, Switzerland, 3010040001). Then, the membrane was blocked in 5% nonfat milk for 1 h at room temperature, incubated with primary antibodies (Appendix A) overnight at 4 °C, washed with TBST, and incubated with horseradish peroxidase-conjugated secondary antibodies (Proteintech, Rosemont, IL, USA, SA00001-1/2) for 1 h at room temperature. An ECL-enhanced chemiluminescent substrate kit (Advansta, K-12045-D50) and Odyssey system (Li-Cor Biosciences, Lincoln, NE, USA) were used for visualization and quantification.

### 4.10. RNA Interference

For gene knockdown, transfections were performed using the control or *OLR1* siRNA (GenePharma, Shanghai, China) in combination with Lipofectamine RNAiMAX reagent (Lipofectamine™ 3000, Invitrogen, Waltham, MA, USA, L3000001), which were suitably diluted in an Opti-MEM medium (Gibco, Billings, MT, USA, 11058021). The resulting mixture was gently introduced into the Cal27 and SCC4 cells (2 × 10^5^), followed by an incubation period of 24 h at 37 °C. The sequences used were as follows: *OLR1* siRNA-1, 5′-CCAGCAAGCAAUUUCCUAUTT-3′; *OLR1* siRNA-2, 5′- CCCTTGCTCGGAAGCTGAATGAGAA-3′; and siCtrl, 5′-UUCUCCGAACGUGUCACGUTT-3′.

### 4.11. Transwell Invasion Assay

The 24-well Transwell chambers (8 um pore, Corning, Corning, NY, USA) were prepared overnight with 60 µL matrix gel (BD Biosciences, NJ, USA, 256234). The transfected HNSCC cells (5 × 10^4^ cells) were inoculated in the upper chamber with 200 µL of serum-free DMEM, and 600 μL of DMEM containing 10% FBS was added to the lower chamber. After 48 h of incubation at 37 °C, the cells were fixed with 4% paraformaldehyde and stained with crystal violet. The uninvaded upper cells were erased. Cell invasion was captured under an inverted microscope.

### 4.12. Wound Healing Assay

The transfected HNSCC cells were inoculated in 6-well plates (1 × 10^6^ cells); when the density reached approximately 95%, the cells were scratched with a 200 µL pipette tip. After cleaning the cell surface with PBS, the medium was replaced with fresh serum-free medium. The healing dynamics of the affected area were subsequently monitored using high-powered microscopic visualization at 0- and 24-hour intervals.

### 4.13. Cell Proliferation

The experiments were performed according to the protocol of our previous study [59]. Briefly, the proliferation rates of the Cal27 and SCC4 cells were measured using the CCK-8 assay (Dojindo, Japan). HNSCC cells at a density of 2000 cells per well in the corresponding medium were inoculated into 96-well plates for the CCK-8 assay. Then, the absorbance was measured at 450 nm using a BioTek plate reader (BioTek, Winooski, VT, USA). 

### 4.14. Sphere Formation Assay

The HNSCC single cells were resuspended in a sphere-forming medium (DMEM/F12 + N2 supplement (1%, R&D Systems, Minneapolis, MN, USA, AR009) + B27 (1%, Gibco, A3582801) +bFGF (20 ng/mL, Invitrogen, RP-8627) + EGF (20 ng/mL, Gibco, AF-100-15-1MG)) and inoculated in ultra-low attachment 6-well plates (1000 cells/well, Corning, Corning, NY, USA). After 10 days of culture, cancer spheres larger than 100 μm in diameter were counted. The sphere formation assay was used to assess the sphere formation ability of the HNSCC cells. 

### 4.15. Flow Cytometry

To detect cell stemness, an ALDEFLUOR^TM^ kit (STEMCELL Technologies, Vancouver, BC, Canada) was used to detect aldehyde dehydrogenase (ALDH) activity as previously described [59]. Briefly, *OLR1* knockdown and control cells were incubated with ALDEFLUOR reagent, and diethylaminobenzaldehyde (DEAB) was used as a negative control. For the cell death detection assay, cells were digested with EDTA-free trypsin and washed with PBS buffer followed by detection with an Annexin V-FITC/PI apoptosis kit (EBioscience, San Diego, CA, USA). The data were acquired using CytoFLEX S (Beckman Coulter, Brea, CA, USA) and analyzed by the FlowJo software (V7.6.2, Treestar, Woodburn, OR, USA). PI-positive cells were counted to assess the occurrence of cuproptosis.

### 4.16. Statistical Analysis

All statistical analyses were performed using the GraphPad Prism 8.0 and R 3.6.3 software. Two-tailed Student’s *t*-test was used to perform comparisons between two groups, and one-way ANOVA and Tukey’s multiple comparison tests were used for multiple comparisons. A Spearman’s correlation analysis was used for the correlation analysis. TIDE scores were compared using the Wilcoxon test. All results were calculated in at least three independent experiments. All error bar values represent standard deviations (SD); *p*-values of <0.05 were considered to be statistically significant (* *p* < 0.05, ** *p* < 0.01, and *** *p* < 0.001).

## 5. Conclusions

In conclusion, we observed that *OLR1* was abnormally upregulated in pan-cancer and was associated with prognostic outcomes. *OLR1* expression was positively correlated with immunosuppressive cell infiltration and immune checkpoint expression in multiple cancers. Additionally, TMB and MSI have also been found to correlate with *OLR1* expression in certain cancers. In HNSCC, the TIDE scores in patients with high *OLR1* expression levels were significantly higher than those for patients with low *OLR1* expression levels. In vitro experiments demonstrated that *OLR1* may affect EMT, invasion, stemness, and proliferation activities via the STAT3 pathway in HNSCC. Furthermore, our novel finding reveals that *OLR1* may prevent cuproptosis by suppressing the oligomerization of DLAT (Figure 9F). Thus, *OLR1* has the potential to serve as a noteworthy biomarker and an emerging therapeutic target for cancer treatment; however, additional research is required to explore its value.

## Figures and Tables

**Figure 1 ijms-24-12904-f001:**
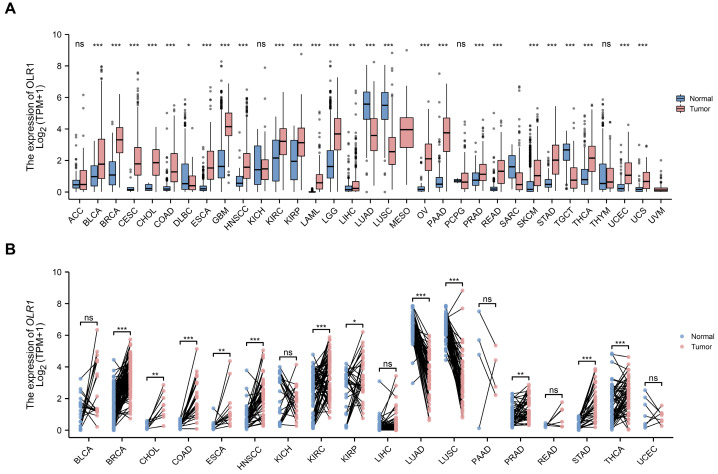
Expression of *OLR1* mRNA in pan-cancer: (**A**) Comparison of *OLR1* expression between tumor and normal samples in 33 cancer types obtained from the TCGA database (* *p* < 0.05; ** *p* < 0.01; *** *p* < 0.001; ns, not significant); (**B**) comparison of *OLR1* expression between tumor and paired normal samples in 18 cancer types obtained from the TCGA database (* *p* < 0.05; ** *p* < 0.01; *** *p* < 0.001; ns, not significant). Data are mean ± s.d.

**Figure 2 ijms-24-12904-f002:**
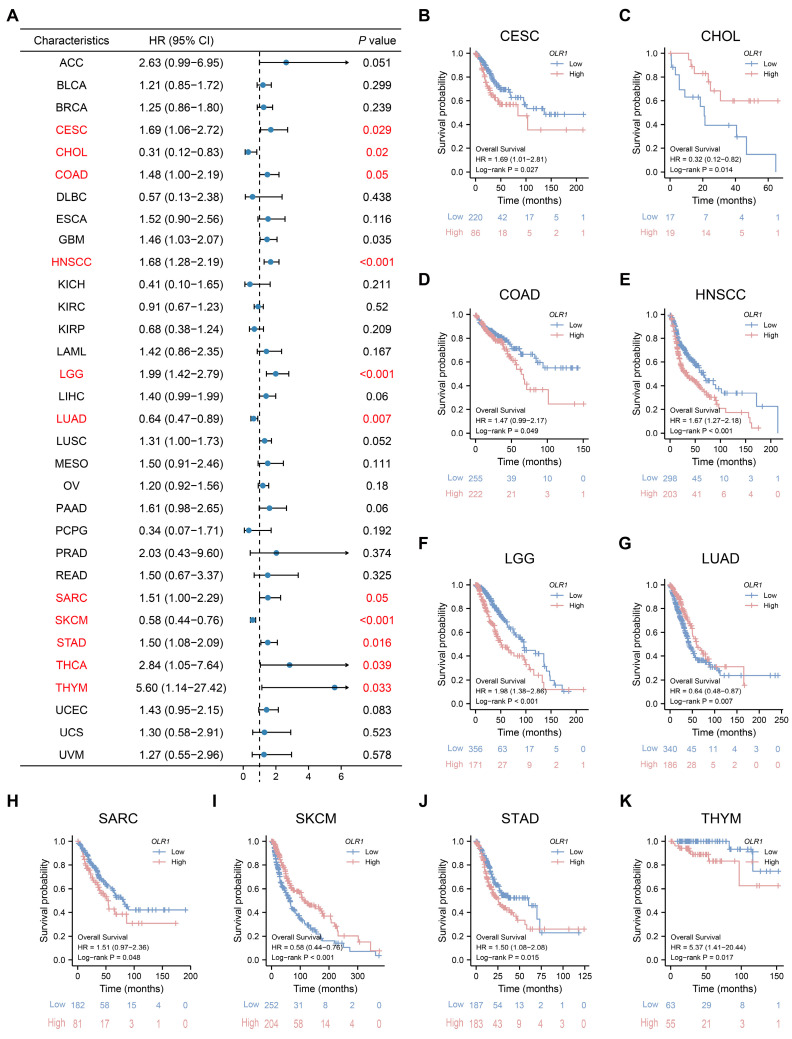
Prognostic analysis of *OLR1* in different cancer types: (**A**) Forest plot showing the correlation between *OLR1* expression and OS in various cancers; (**B**–**K**) Kaplan–Meier survival analysis results show that overall survival is decreased in CESC, COAD, HNSCC, LGG, SARC, STAD, and THYM patients with higher *OLR1* expression levels, whereas the opposite result is observed in CHOL, LUAD, and SKCM patients.

**Figure 3 ijms-24-12904-f003:**
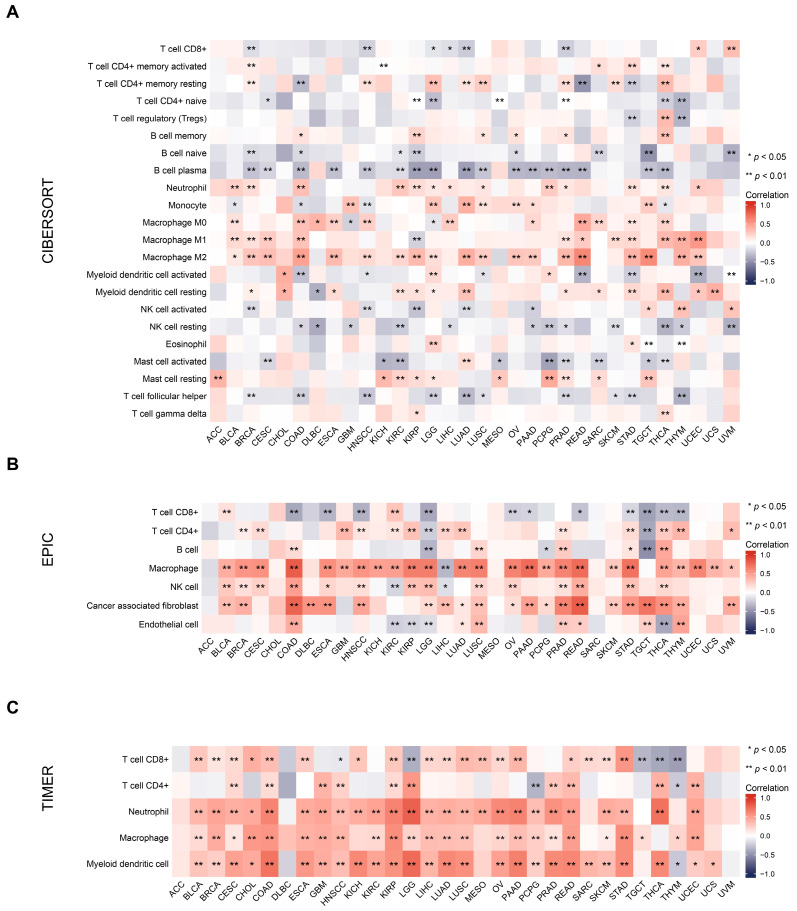
Correlation between *OLR1* expression and immune cell infiltration in various pan-cancer types using: (**A**) The CIBERSORT algorithm; (**B**) the EPIC algorithm; (**C**) the TIMER algorithm.

**Figure 4 ijms-24-12904-f004:**
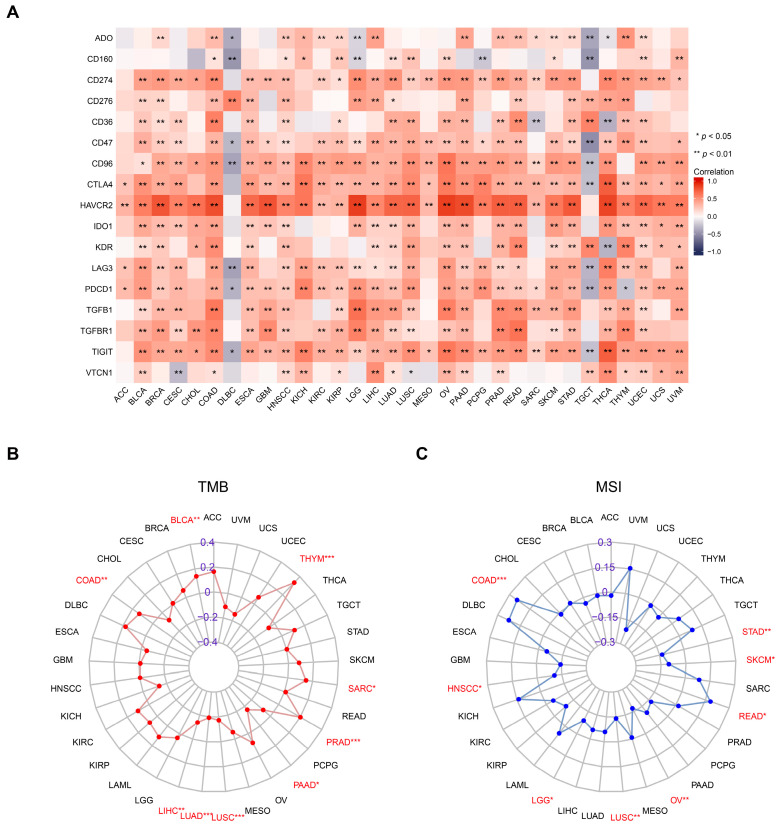
Overexpressed *OLR1* is positively correlated with immune checkpoint molecules in multiple cancers: (**A**) Heatmap showing correlations between *OLR1* expression and 17 common immune checkpoint genes’ RNA levels in the TCGA pan-cancer database by the Spearman’s correlation test; (**B**) radar plot showing the correlation between *OLR1* expression and TMB in different cancers; (**C**) radar plot showing the correlation between *OLR1* expression and MSI in different cancers. * *p* < 0.05, ** *p* < 0.01, and *** *p* < 0.001.

**Figure 5 ijms-24-12904-f005:**
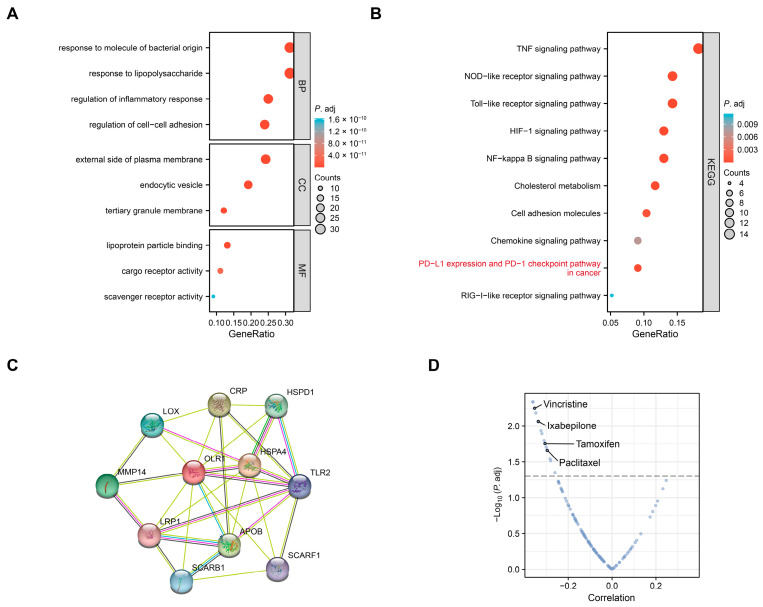
Functional enrichment analysis of *OLR1*-related genes: (**A**) GO and (**B**) KEGG enrichment analyses of the top 100 *OLR1*-related genes; (**C**) protein–protein interaction network for *OLR1* using the STRING datase; (**D**) drug sensitivity analysis of *OLR1* using the CellMiner database. Dashed line indicates *p* < 0.05.

**Figure 6 ijms-24-12904-f006:**
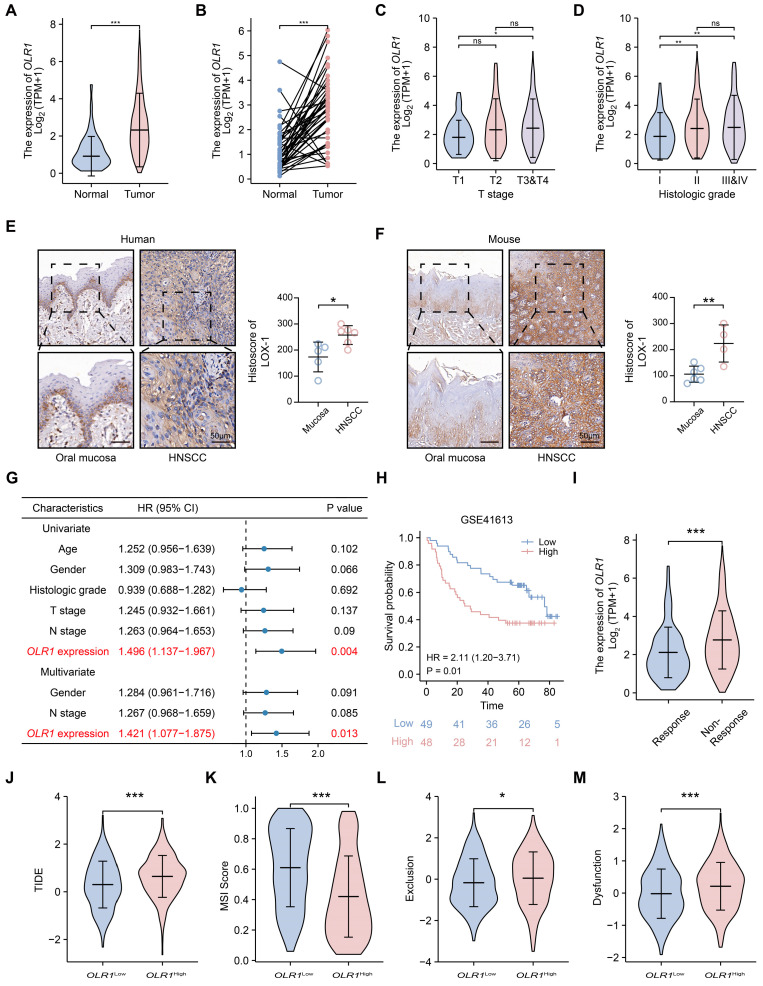
Increased *OLR1* expression is associated with poor prognosis and ICI resistance in HNSCC: (**A**) Violin plot showing that *OLR1* expression is higher in tumors (*n* = 502) than in normal controls (*n* = 44) from the TCGA HNSCC cohort (*** *p* < 0.001); (**B**) *OLR1* was upregulated in tumors (*n* = 43) compared to paired adjacent controls (*n* = 43) (*** *p* < 0.001); (**C**) violin plot showing *OLR1* expression in different tumor T stages and (**D**) grades of HNSCC ( * *p* < 0.05, ** *p* < 0.01, and ns, not significant); (**E**) representative IHC images and quantification of LOX-1 in human mucosa (*n* = 5) and HNSCC (*n* = 6) (* *p* < 0.05 and bar, 50 μm); (**F**) representative IHC images and quantification of LOX-1 in mouse mucosa (*n* = 6) and HNSCC (*n* = 4) (** *p* < 0.01 and bar, 50 μm); (**G**) *OLR1* expression was an independent poor prognostic factor for HNSCC patients according to the Cox regression analysis; (**H**) Kaplan–Meier analysis showed that HNSCC patients with high *OLR1* expression levels had worse OS rates in the GSE41613 dataset (median cut-off, *n* = 97) (**I**) violin plot showing that *OLR1* expression is higher in ICI-resistant HNSCC patients (*n* = 349) than in ICI-responsive HNSCC patients (*n* = 153) (*** *p* < 0.001); (**J**) TIDE; (**K**) MSI, (**L**) T-cell exclusion, and (**M**) dysfunction scores compared between the different *OLR1* subgroups using the Wilcoxon test (* *p* < 0.05, *** *p* < 0.001). Data are mean  ±  s.d.

**Figure 7 ijms-24-12904-f007:**
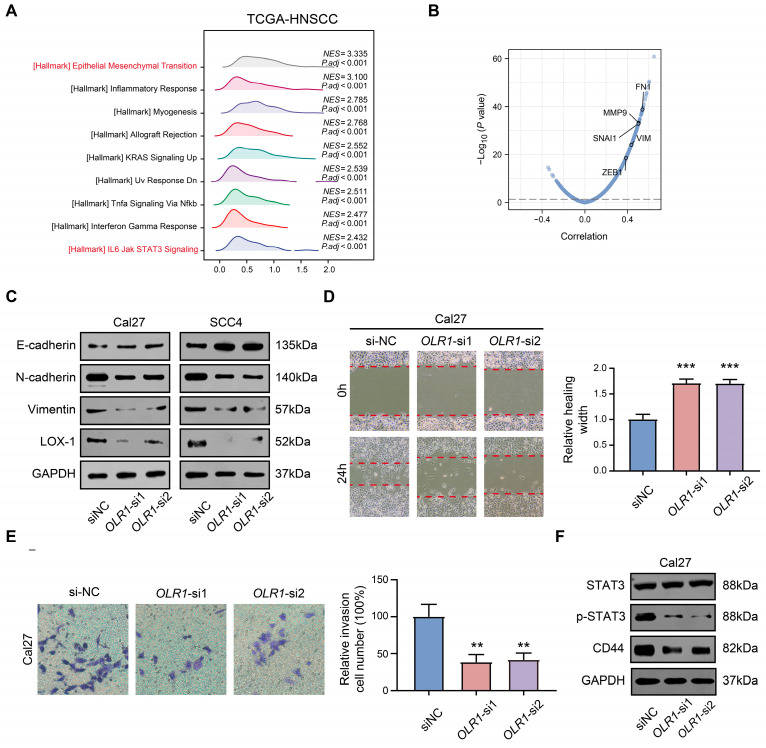
Effect of *OLR1* knockdown on the EMT process in HNSCC cell lines: (**A**) GSEA results of *OLR1* based on the TCGA-HNSCC dataset; (**B**) Volcano plot of *OLR1*-associated genes in the TCGA HNSCC cohort by the Pearson’s correlation coefficient test; (**C**) Western blot analysis of LOX-1, N-cadherin, E-cadherin, and vimentin expressions in *OLR1* knockdown or control Cal27 and SCC4 cells; (**D**) representative images and quantitative analyses of wound healing (20×) and (**E**) Transwell assay (40×) in Cal27 cells (** *p* < 0.01 and *** *p* < 0.001, the red dash indicates the edge of the scratch); (**F**) expression of STAT3, p-STAT3, and CD44 were analyzed by Western blot.

**Figure 8 ijms-24-12904-f008:**
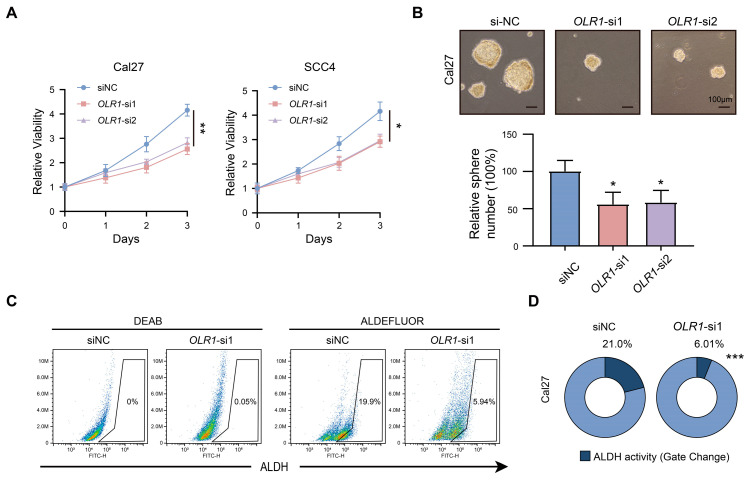
*OLR1* participates in the maintenance of stemness characteristics in HNSCC cells: (**A**) Effect of *OLR1* knockdown on Cal27 and SCC4 cell growth as determined by CCK-8 (* *p* < 0.05 and ** *p* < 0.01); (**B**) the effect of *OLR1* knockdown on the stemness properties of Cal27 cells was evaluated using the sphere formation assay (* *p* < 0.05 and bar, 100 μm), data are mean  ±  s.d.; (**C**,**D**) ALDH activity in *OLR1*-deficient and control cells was quantified by flow cytometry, *** *p* < 0.001.

**Figure 9 ijms-24-12904-f009:**
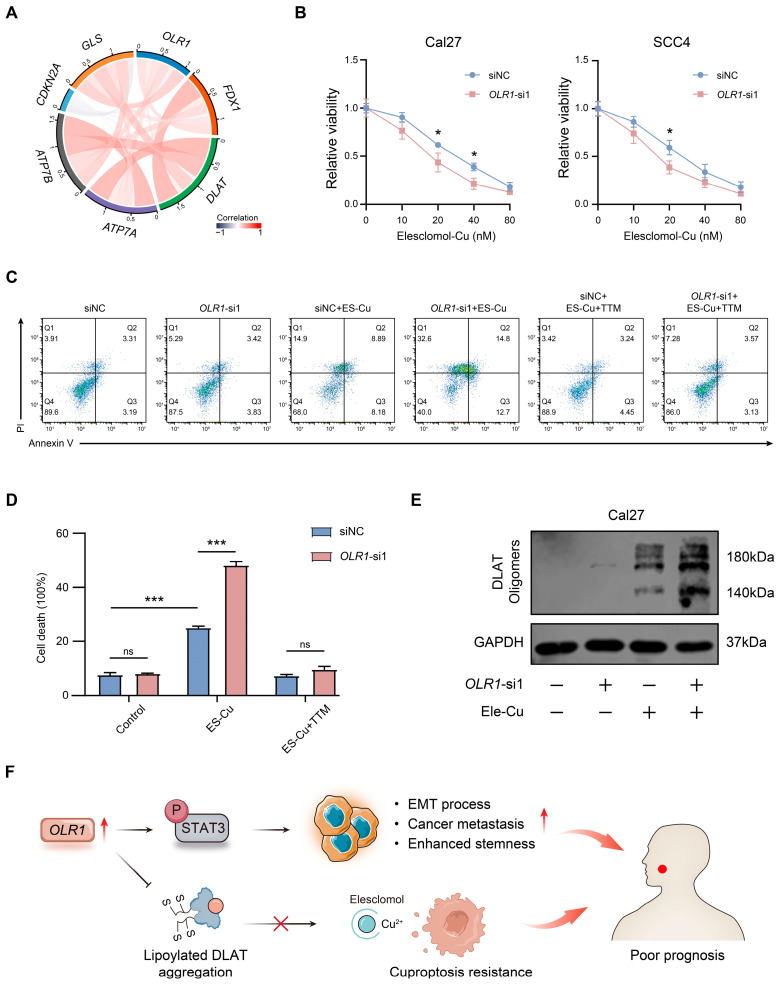
*OLR1* acts as a repressor of cuproptosis in HNSCC cells: (**A**) Correlations of *OLR1* and cuproptosis-related genes in the TCGA-HNSCC dataset; (**B**) cell viability of Cal27 and SCC4 cells following treatment with the indicated concentrations of elesclomol (* *p* < 0.05); (**C**) cuproptosis cells were stained with Annexin V and PI for flow cytometry in Cal27 treated with elesclomol (40 nM) and TTM (20 nM); (**D**) quantitative statistical analysis shows different percentages of cell death in si*OLR1* and the control group (*** *p* < 0.001, ns, no significant difference); (**E**) DLAT oligomerization was analyzed by Western blot in Cal27 cells that were treated with elesclomol (40 nM); (**F**) schematic illustration of the role of *OLR1* in HNSCC. In brief, the elevated expression of *OLR1* activates the STAT3 signaling pathway to regulate metastasis, the EMT process, and stemness characteristics in HNSCC. In addition, *OLR1* negatively modulates cuproptosis by inhibiting DLAT oligomerization. In summary, *OLR1* is associated with poor prognosis and may serve as an emerging target for cancer therapy.

## Data Availability

The original data presented in this study are included in the article or Appendix A; additional information can be requested from the corresponding authors.

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
