# Peer review of "OLR1 Is a Pan-Cancer Prognostic and Immunotherapeutic Predictor Associated with EMT and Cuproptosis in HNSCC"

_ijms, 2023, doi:10.3390/ijms241612904_

Round 1

Reviewer 1 Report

In the present manuscript, Wu et al. assess expression and prognostic implications of the Oxidized low-density lipoprotein receptor 1 (OLR1) gene in a combined in silico and in vitro analysis in several cancers, drawing an association with a potential more pro-immunogenic tumor phenotype. In HNSCC, authors reproduce and deepen this hypothesis in vitro, proposing OLR1 as a potential promising future immune-related prognostic biomarker.

Overall, the manuscript provides a comprehensive analysis of the potential role of OLR1, building a credible hypothesis basing on TCGA-data, which the authors conclusively discuss using HNSCC data. Further clinical validation appears promising. Considering the novelty and potential clinical applicability of herein presented data, I may hereby recommend the manuscript for publishing, given the authors consider following remarks:

Introduction

The introduction is well-written, allows to follow a thorough red thread and adequately  introduces readers, who are not familiar with OLR1, into the matter.

However, the last paragraph (lines 68-81) is misplaced. No summary of results is supposed to be given within the introduction. I suggest to remove lines 68- 81 completely and replace it by outlining the study aims here, like e.g. the present study therefore aims to evaluate…

Methods

Methods are described comprehensively; however, details are sometimes provided only very briefly.

Line 118 – please quote the previous study and give details on clinical details (primary? Recurrence? Stage? – as these details may deeply influence a tumors immunogenicity) and how the samples were retrieved. In case I have overseen this data, I may ask the authors to provide it in a supplementary table.

2.6. Could you please explain your choice of immunstaining techniques? Line 124 “..followed by incubation with secondary antibodies (ZSGB-BIO)” – please provide all detailed information on all used antibodies, including manufacturer, catalogue number and dilution to secure reproducibility.

2.9. Linge 143f: Again, please provide manufacturer details and catalogue numbers.

Results and Discussion

3.6 cuproptosis:

Lines 389-397 “recent studies… decided to explore” – this paragraph is completely misplaced in the results section. To me, it sounds like as if the authors post-hoc reconsidered their methods and added the cuproptosis part during the analysis, which was not part of the initial study protocol. I may suggest to explain the association between OLR1 and cuproptosis in the introduction, give the methodology in the methods section and only present the results in the results section. Again, line 406-408, interpreting own results is not part of the results section, but supposed to be part of the discussion section. Please thoroughly revise this paragraph.

Once again, lines 490-503, the discussion around cuproptosis appears like an appendix which was added after the manuscript was finished and does not really fit the flow of the discussion. How do authors interpret their findings in the light of existing literature? Is previous evidence on an association of cuprpptosis and OLR1 for HNSCC available? Could it be related to the clinically potentially valuable associations with a more pro-immunogenic phenotype?

Finally, I miss a section discussing the flaws and limitations of the present paper. Please add a respective paragraph.

General remark: Please provide your figures in an adequate resolution. E.g., Figures 6E, 6F or 8B are blurry and do not allow to see details relevant to follow the author’s points.

Reviewer 2 Report

Wu et al explored OLR1 metabolic gene and its expression in cancer it is interesting study. I have following major concerns

1)    In section 3.1 to understand the OLR1 expression in cancer authors used TCGA database but how exactly it Is done ? method section related to this analysis is completely missing.

2)    In section 3.2 authors explored OLR1 expression in relation to immune system.

What is the motivation to do this OLR1 is a metabolic gene its correlation to immune system is not clear. Again method section related to this missing.

3)    In section 3.3 authors constructed protein interaction network of OLR1 and included top 30 proteins why only 30? I would suggest to construct comprehensive network include all interactions and analyze their pathway. And include all details in methods.

4 ) In section 3.4 authors chose HNSCC cancer for biological validation. It is not clear why this cancer was choosen for validation

Round 2

Reviewer 1 Report

I thank the authors for the constructive revisions. I may recommend to accept the manuscript in present form.

Reviewer 2 Report

Authors responded to all my comments